# Low-Level Tolerance to Fluoroquinolone Antibiotic Ciprofloxacin in QAC-Adapted Subpopulations of *Listeria monocytogenes*

**DOI:** 10.3390/microorganisms9051052

**Published:** 2021-05-13

**Authors:** Divya Kode, Ramakrishna Nannapaneni, Mohit Bansal, Sam Chang, Wen-Hsing Cheng, Chander S. Sharma, Aaron Kiess

**Affiliations:** 1Department of Food Science, Nutrition and Health Promotion, Mississippi State University, Starkville, MS 39762, USA; dsk125@msstate.edu (D.K.); schang@fsnhp.msstate.edu (S.C.); wcheng@fsnhp.msstate.edu (W.-H.C.); 2Poultry Science Department, Mississippi State University, Starkville, MS 39762, USA; mb3405@msstate.edu (M.B.); csharma@poultry.msstate.edu (C.S.S.); akiess@poultry.msstate.edu (A.K.)

**Keywords:** *Listeria monocytogenes*, QAC, sublethal adaptation, biocides, ciprofloxacin, antibiotics

## Abstract

There was a development of low-level tolerance to fluoroquinolone antibiotic ciprofloxacin in *Listeria monocytogenes* after sublethal adaptation to quaternary ammonium compound (QAC). Using eight *L. monocytogenes* strains, we determined the changes in short-range MIC, growth rate, and survival for heterologous stress response to ciprofloxacin, after sublethal exposure to daily cycles of fixed or gradually increasing concentration of QAC. Three main findings were observed. (1) MIC increase—QAC-adapted subpopulations exhibited a significant increase in short-range MIC of ciprofloxacin, by 1.5 to 2.9 fold, as compared to non-adapted control for 4/8 strains (*p* < 0.05). (2) Growth rate increase—QAC-adapted subpopulations exhibited significant 2.1- to 6.8- fold increase in growth rate (OD_600_ at 10 h) in ciprofloxacin-containing broth, as compared to non-adapted control for 5/8 strains (*p* < 0.05). (3) Survival increase—QAC-adapted subpopulations of *L. monocytogenes* yielded significantly higher survival in ciprofloxacin-containing agar by 2.2 to 4.3 log CFU/mL for 4/8 strains, as compared to non-adapted control (*p* ˂ 0.05). However, for other 4/8 strains of *L. monocytogenes*, there was no increase in survival of QAC-adapted subpopulations, as compared to non-adapted control in ciprofloxacin. These findings suggest the potential formation of low-level ciprofloxacin-tolerant subpopulations in some *L. monocytogenes* strains when exposed to residual QAC concentrations (where QAC might be used widely) and such cells if not inactivated might create food safety risk.

## 1. Introduction

Recent findings indicate that there is a slow emergence of antibiotic resistance in *L. monocytogenes*. The first antibiotic-resistant *L. monocytogenes* was isolated in 1988 from a clinical sample in France [1]. About twelve years later in Florida and the Greater Washington DC in 2002–2003, a majority of the 167 isolates of *L. monocytogenes* obtained from retail foods such as ready-to-eat (RTE) meats, raw chicken carcasses, and fresh produce, were found to be resistant to three antibiotics—ciprofloxacin, tetracycline, and sulfonamide [2]. Then in Italy in 2009, a multidrug-resistant *L. monocytogenes* was isolated from food and food processing environments, which was resistant against five antibiotics, including, ciprofloxacin, moxifloxacin, linezolid, vancomycin, and trimethoprim/sulfamethoxazole, and with intermediate resistance to clindamycin [3]. Additionally, in Spain, between 1993 and 2006, a significant increase in resistance was reported in *L. monocytogenes* strains isolated from poultry houses for six antibiotics, including, gentamicin, streptomycin, neomycin, enrofloxacin, ciprofloxacin, and furazolidone [4]. Recently in China, an increase in resistance was reported in *L. monocytogenes* from meat and meat product samples, which exhibited multidrug resistance against five antibiotics, including, tetracycline, penicillin G, streptomycin, ampicillin, and cephalothin [5].

Biocides or disinfectants are essential for eliminating or reducing the pathogenic and spoilage microorganisms from food-contact surfaces, equipment, walls, and floors. Although chemical disinfectants are extensively used in the food industry to reduce contaminating microbial load and to maintain hygiene standards, they might fail to eradicate bacteria and could potentially result in bacterial survival. Quaternary ammonium compounds (QACs) are a class of disinfectants that are widely used in the healthcare and food processing industries. QACs are cationic membrane-active agents that interact with the cytoplasmic membrane of the microbial cell, along with intracellular entities like DNA. Their classification is based on the nature of the ‘R’ group, which causes structural variations that attribute to their antimicrobial activity and are extremely potent when used at manufacturers’ recommended concentrations [6]. Many factors are responsible for reducing QAC efficacy, which then results in sublethal concentrations. The biocidal activity is dependent on temperature, pH, concentration, contact time, and type and load of organic material, and any changes in these parameters significantly affect the bactericidal potency [7]. The presence of organic matter has a significant effect on the efficacy of biocide as it might create a physical hindrance and protect the bacteria against disinfectant action [8]. QACs are biodegradable under aerobic conditions and hence their concentrations in the environment vary considerably [9]. In addition, the application of QACs on wet surfaces dilutes the original lethal concentrations, promoting conditions for sublethal or gradient concentrations [10]. Generally, dilutions occur due to the accumulation of water in places that are assumed to be dry, or in hollow equipment, crevices, cracks, and floors [11].

Despite the frequent cleaning and sanitation sequence, strains of *L. monocytogenes* are recurrently isolated from food industry premises, such as refrigerated rooms, floors, drains, and equipment, due to the innate ability of this pathogen to adapt to stresses encountered, leading to its continuous persistence [12,13,14,15]. Persistent *L. monocytogenes* strains were frequently isolated from food processing plants that exhibited a low-level resistance to QAC [16]. Several studies also indicate that the use of QACs impose selective pressure, contributing to the emergence of antibiotic-resistant strains. For example, sublethal exposure to progressively increasing concentrations of benzalkonium chloride (a QAC formulation) in *L. monocytogenes*, resulted in the reduced susceptibility to gentamicin and kanamycin [17]. A tetracycline-resistant *L. monocytogenes* strain was isolated following a stepwise exposure to a QAC-based disinfectant, didecyl dimethyl ammonium chloride (DDAC) [18]. The persistent strains of *L. monocytogenes* isolated from pig slaughterhouses exhibited resistance against ampicillin, penicillin, and trimethoprim/sulfamethoxazole [19].

Ciprofloxacin belongs to the fluoroquinolone family of antibiotics, which are extensively used in human medicine. Recently, reduced susceptibility towards fluoroquinolone was reported in *L. monocytogenes* strains [20]. The derivatives of *L. monocytogenes* isolated after QAC adaptation showed reduced susceptibility to ciprofloxacin [21]. The exposure to progressively increasing concentrations of QAC resulted in some strains exhibiting increased resistance to ciprofloxacin [22]. Despite these studies, our understanding of the role of QAC and the mechanisms underlying the heterologous stress-response in *L. monocytogenes* is still limited. Often the methods used for monitoring changes such as minimum inhibitory concentration (MIC) rely on endpoint measurements, and do not take into account the growth kinetics of the microorganism. In the presence of antibiotics, the growth profile of the microorganisms is complex. Therefore, monitoring the growth kinetics is an appropriate tool for determining dynamic bacterial growth patterns in the presence of antibiotics [23]. Therefore, in the present study, we investigated the heterologous stress-response to ciprofloxacin in *L. monocytogenes* strains, after sublethal adaptation to fixed or gradually increasing concentration of QAC with the following three objectives. (1) Determine the changes in short-range MIC of ciprofloxacin for QAC-adapted subpopulations of *L. monocytogenes*. (2) Determine the growth kinetics of QAC-adapted subpopulations of *L. monocytogenes* in ciprofloxacin-containing broth model, in comparison to the non-adapted control cells. (3) Determine the survival of QAC-adapted subpopulations of *L. monocytogenes* in ciprofloxacin-containing agar model versus non-adapted control cells. These findings are useful in understanding the potential link between biocide tolerance and the emergence of fluoroquinolone-resistant strains of *L. monocytogenes*, in food processing environments that might increase food safety risk.

## 2. Materials and Methods

### 2.1. Listeria monocytogenes Strains and Growth Conditions

*L. monocytogenes* strains used in this study are listed in Table 1. *L. monocytogenes* strains were stored at −20 °C in Tryptic soy broth containing 0.6% yeast extract (TSBYE) (BD Bio sciences, San Jose, CA, USA) and were supplemented with 25% glycerol. The frozen stocks were subcultured on PALCAM agar at 37 °C for 24 h. To prepare the working stock cultures, a single colony from PALCAM agar was inoculated into 10 mL TSBYE and incubated at 37 °C for 18–24 h. Overnight cultures were prepared for each experiment by inoculating a loopful from the working stock into 10 mL TSBYE kept at 37 °C for 18–24 h, to reach an inoculum level of approximately 9 log CFU/mL.

### 2.2. Preparation of QAC Solutions

The 50% benzalkonium chloride (ACROS Organics™, Fair Lawn, NJ, USA) was diluted in sterile distilled water to prepare a stock of 5000 µg/mL. Then, working stocks of 1 to 8 µg/mL were prepared in TSBYE or in sterile distilled water before each experiment, by diluting the 5000 µg/mL solution.

### 2.3. L. monocytogenes Adaptation to Sublethal QAC

Three methods (two broth models and one water model) were used for the daily cycles of sublethal QAC exposure against planktonic cells of *L. monocytogenes* in 24-well plates (Techno Plastic Products, AG, Switzerland). (1) A fixed concentration of QAC in the broth model (QAC-P1); (2) a gradually increasing concentration of QAC in broth model (QAC-P2); and (3) a gradually increasing concentration of QAC in distilled water model (QAC-P3). The complete protocols are explained below.

### 2.4. Preparation of QAC-Adapted Subpopulation 1 in Broth Model (QAC-P1)

For QAC-P1, *L. monocytogenes* strains were exposed to a fixed sublethal concentration of QAC at 2 µg/mL in daily overnight cycles for 5 days in the broth model. In brief, 100 µL aliquot of overnight grown cells diluted to 10^8^ CFU/mL was added to 900 µL of TSBYE in 24-microtiter wells, to yield a 1 mL of 10^7^ CFU/mL inoculum. Then, 200 µL TSBYE containing 4 µg/mL QAC was added at each hourly interval for five hours, to yield a final QAC concentration of 2 µg/mL in a final volume of 2 mL/well for overnight incubation. The 24-well plate was then incubated at room temperature (22 °C) for 19 h until it reached 10^9^ CFU/mL or max OD_600_ (>0.9). The newly QAC- adapted cells were then used to re-run the above daily cycle for 5 days.

### 2.5. Preparation of QAC-Adapted Subpopulation 2 in Broth Model (QAC-P2)

For QAC-P2, *L. monocytogenes* strains were exposed to a gradually increasing concentration of QAC from 0.5 µg/mL to 4 µg/mL, in daily overnight cycles, for 8 days in the broth model. In brief, 100 µL aliquot of overnight grown cells diluted to 10^8^ CFU/mL was added to 900 µL of TSBYE in 24-microtiter wells to give rise to a 1 mL of 10^7^ CFU/mL inoculum. Then, 200 µL TSBYE containing QAC was added at each hourly interval for five hours to yield a final QAC concentration of 0.5 µg/mL (day 1), 1 µg/mL (day 2), 1.5 µg/mL (day 3), 2 µg/mL (day 4), 2.5 µg/mL (day 5), 3 µg/mL (day 6), 3.5 µg/mL (day 7), or 4 µg/mL (day 8) in a final volume of 2 mL/well, before overnight incubation. The 24-well plate was then incubated at room temperature (22 °C) for 19 h until it reached 10^9^ CFU/mL or OD_600_. The newly QAC-adapted cells were then used to re-run the next daily cycle for 8 days, as described above.

### 2.6. Preparation of QAC-Adapted Subpopulation 3 in Water Model (QAC-P3)

For QAC-P3, *L. monocytogenes* strains were exposed to a gradually increasing concentration from 0.5 µg/mL to 4 µg/mL in daily cycles, for 8 days in the water model. In brief, 100 µL aliquot of overnight grown cells diluted in sterile distilled water to 10^8^ CFU/mL was added to 100 µL sterile distilled water containing QAC, to yield a final concentration of 0.5 µg/mL (day 1), 1 µg/mL (day 2), 1.5 µg/mL (day 3), 2 µg/mL (day 4), 2.5 µg/mL (day 5), 3 µg/mL (day 6), 3.5 µg/mL (day 7), or 4 µg/mL (day 8), for 1 h, and then supplemented with 1.8 mL of TSBYE containing 0.5 µg/mL QAC, to yield a final volume of 2 mL/well before overnight incubation. The 24-well plate was then incubated at room temperature (22 °C) for 23 h until it reached 10^9^ CFU/mL or OD_600_. The new QAC-adapted cells were then used to re-run the next daily cycle for 8 days, as described above.

### 2.7. Preparation of Ciprofloxacin Solutions

The stock solution of ciprofloxacin (Ciprofloxacin, 98%, ACROS Organics™, Fair Lawn, NJ, USA) at 500 µg/mL was prepared in sterilized distilled water and stored at 4 °C for 2 weeks. Prior to each experiment, working stock solutions of 1 to 10 µg/mL were prepared in TSBYE from the 500 µg/mL stock.

### 2.8. Determination of Short-Range MIC of Ciprofloxacin for QAC-Adapted Subpopulations and Non-Adapted Control of L. monocytogenes Strains

The changes in susceptibilities of QAC-adapted subpopulations and non-adapted control of *L. monocytogenes* strains against ciprofloxacin were determined using the short-range minimum inhibitory concentration (MIC) method described previously [10], with slight modifications. The ciprofloxacin concentrations in the range of 1 to 10 µg/mL with 1 µg/mL increments were tested in the short-range MIC. The overnight grown QAC-adapted subpopulations and non-adapted control were diluted to 10^7^ CFU/mL in TSBYE and an aliquot of 30 µL was inoculated into individual wells of a 96-well microtiter plate containing 270 µL of ciprofloxacin concentrations range, such that the resulting final inoculum concentration in the wells was 10^6^ CFU/mL. The MIC was defined as the lowest concentration of ciprofloxacin preventing the growth, after incubation at 37 °C for 48 h. Optical density at 600 nm (OD_600_) was recorded after 48 h.

### 2.9. Determination of Growth Kinetics of QAC-Adapted Subpopulations and Non-Adapted Control of L. monocytogenes Strains in Ciprofloxacin-Containing Broth

Growth kinetics of QAC-adapted subpopulations and non-adapted control of *L. monocytogenes* strains were evaluated in the presence of sublethal concentrations of ciprofloxacin in TSBYE. The overnight grown QAC-adapted and non-adapted control cells (10^9^ CFU/mL) were diluted to 10^7^ CFU/mL in TSBYE. The growth response of *L. monocytogenes* strains was monitored in TSBYE containing 2 µg/mL ciprofloxacin concentration. In brief, 30 µL aliquot of 10^7^ CFU/mL was added to 270 µL of TSBYE containing 2 µg/mL ciprofloxacin in a 96-well polystyrene plate, and incubated at 37 °C in a shaker at 150 rpm (C24 Classic series incubator shaker, New Brunswick Scientific Inc., Edison, NJ). Growth was monitored by recording OD_600_ at every 4 h for up to 24 h (ELx800 Absorbance Microplate Reader, BioTek Instruments, Inc., Winooski, VT, USA).

### 2.10. Determination of Survival of QAC-Adapted Subpopulations and Non-Adapted Control of L. monocytogenes Strains in Ciprofloxacin-Containing Agar

The survival of QAC-adapted subpopulations and non-adapted control of *L. monocytogenes* strains were determined on Tryptic soy agar containing 0.6% yeast extract (TSAYE) supplemented with 2 µg/mL ciprofloxacin. In brief, overnight grown QAC-adapted and non-adapted cells of *L. monocytogenes* strains were diluted to 10^7^ CFU/mL. Subsequently, four decimal dilutions were performed and 50 µL of each dilution was spotted twice on TSAYE containing ciprofloxacin and TSAYE without ciprofloxacin (control plate). The plates were incubated at 37 °C to determine CFU counts after 48 h.

### 2.11. Statistical Analysis

A completely randomized design with a 2 × 3 factorial structure (adapted cells and non-adapted control cells vs. three QAC-adapted subpopulations) in a randomized complete block design, with replication considered as a block. The ciprofloxacin MIC between QAC-adapted subpopulations and non-adapted control were compared using the unpaired two-tailed *t*-test at three significance levels (*p* < 0.05; *p* < 0.01; and *p* < 0.001) using Microsoft Excel (Microsoft Excel, Version 2008). The lag phase duration at OD_600_ between QAC-adapted subpopulations and non-adapted control were compared using the unpaired two-tailed *t*-test at three significance levels (*p* < 0.05; *p* < 0.01; and *p* < 0.001), using Microsoft Excel (Microsoft Excel, Version 2008). The fold increase and percentage increase in ciprofloxacin MIC and fold increase in growth rate was tested by Duncan’s multiple range test (*p* < 0.05) using the SAS software (SAS 9.4 TS Level 1M5; SAS Institute Inc., Cary, NC, USA). Statistical analysis was performed using one-way ANOVA at a significance level of *p* < 0.05. Bacterial counts from each replicate were log-transformed in Microsoft excel before statistical analysis. Counts for survival were analyzed using one-way ANOVA in a completely randomized block design and the means were separated by Fisher’s protected LSD when *p* < 0.05 using the SAS software (SAS 9.4 TS Level 1M5; SAS Institute Inc., Cary, NC, USA).

## 3. Results

### 3.1. Changes in Short-Range MIC of Ciprofloxacin for QAC-Adapted Subpopulations of L. monocytogenes

Figure 1 shows the changes in short-range MIC of ciprofloxacin for eight *L. monocytogenes* strains before and after sublethal exposure to daily cycles of fixed QAC concentration in broth model (QAC-P1), or gradually increasing QAC concentration in broth model (QAC-P2), or gradually increasing QAC concentration in water model (QAC-P3). Table 2 shows the fold increase in short-range MIC of three QAC-adapted subpopulations versus non-adapted control for eight *L. monocytogenes* strains.

Before sublethal QAC exposure, *L. monocytogenes* strains exhibited MICs in the range of 2.3 to 6.0 µg/mL ciprofloxacin. After sublethal QAC exposure, the short-range MIC of ciprofloxacin for all strains of *L. monocytogenes* increased significantly to the range of 5.7 to 8.0 µg/mL for QAC-P1, 4.0 to 8.0 µg/mL for QAC-P2, and 4.3 to 7.7 µg/mL for QAC-P3 (Figure 1). This increase in the short-range MIC of ciprofloxacin by 2.6 to 3.3 µg/mL of QAC-P1, or 2.0 to 4.3 µg/mL of QAC-P2, or 2.3 to 3.7 µg/mL of QAC-P3 was found to be significant (*p* < 0.01 or *p* < 0.05) for 4/8 *L. monocytogenes* strains, as compared to non-adapted control (Figure 1C,E–G). Additionally, for these 4/8 *L. monocytogenes* strains (ScottA, NRRL B 33155, NRRL B 33157, and ATCC 43257), this change in short-range ciprofloxacin MIC was equivalent to 1.6 to 2.6 fold (=60–157%) higher for QAC-P1, or 1.5 to 2.9 fold (=46–186%) higher for QAC-P2, or 1.5 to 2.6 fold (=54–157%) higher for QAC-P3, as a result of the QAC sublethal exposure, which was significant as compared to non-adapted control (*p* < 0.05). On the other end, for 1/8 *L. monocytogenes* strains (ATCC 19116), there was a 1.2 to 1.4 fold (=16–35%) higher in short-range MIC of ciprofloxacin for QAC-P1 or QAC-P2 or QAC-P3 (Table 2).

There were significant strain × subpopulation interactions in the short-range MIC of ciprofloxacin for some *L. monocytogenes* strains. For example, QAC-P1 of strain EGD showed a 1.9 fold increase (=90% higher), as compared to QAC-P2/QAC-P2 with a 1.3 to 1.4 fold increase (=17–40% higher) in the short-range MIC of ciprofloxacin, in comparison to the non-adapted controls. ScottA exhibited the highest fold increase of 2.6 to 2.9 (=157–186% higher) in MIC of ciprofloxacin for QAC-P1, QAC-P2, and QAC-P3, as compared to the non-adapted control (Table 2).

### 3.2. Changes in the Growth Rate of QAC-Adapted Subpopulations of L. monocytogenes in Ciprofloxacin-Containing Broth

The growth curves (OD_600_) of eight *L. monocytogenes* strains in ciprofloxacin before and after sublethal exposure to fixed or gradually increasing concentration of QAC are shown in the Appendix A for QAC-P1, in Appendix A for QAC-P2, and in Appendix A for QAC-P3. Figure 2 shows the changes in lag phase duration, and Figure 3 shows the increase in growth rate at 10 h for three QAC-adapted subpopulations versus non-adapted control for eight *L. monocytogenes* strains. Table 2 shows the fold changes in growth rate at different time-points for three QAC-adapted subpopulations versus the non-adapted control for eight *L. monocytogenes* strains.

For 5/8 *L. monocytogenes* strains (EGD, ScottA, NRRL B 33155, NRRL B 33157, and ATCC 43527), there was a significant decrease in the lag phase down to 4–5.3 h for QAC-P1 (Appendix A), 3.3–6 h for QAC-P2 (Appendix A), or 4–10.7 h for QAC-P3 (Appendix A), as compared to a longer lag phase of 10–21 h for the non-adapted control in ciprofloxacin-containing broth (*p* < 0.05) (Figure 2A,C,E–G). For these 5/8 strains of *L. monocytogenes*, a significant increase in growth rate (OD_600_) of QAC-P1, QAC-P2, and QAC-P3 was observed in ciprofloxacin-containing broth at the 10-h time-point (Figure 3), which was 3.6 to 5.0 fold higher for QAC-P1, 3.3 to 6.8 fold higher for QAC-P2, or 1.2 to 6.4 fold higher for QAC-P3 versus the non-adapted control (*p* < 0.05) (Table 2). Such an increase in growth rate also continued for the QAC-adapted subpopulations to the 12-h time-point for these 5/8 strains in ciprofloxacin. With few exceptions, the highest increase in growth rate in ciprofloxacin was for the QAC-adapted subpopulations of strain ATCC 43527 or NRRL B 33157 (serotype 4b), followed by EGD (serotype 1/2a) at the 10- or 12-h time-point. At the 20-h time-point in ciprofloxacin, the growth rate of QAC-P1 or QAC-P2 or QAC-P3 for 3/8 *L. monocytogenes* strains (EGD, ScottA, and ATCC 43257) continued to be 2.0 to 5.1 fold higher, as compared to the non-adapted control. At the 24-h time-point, there was a continued 2.1- to 2.3-fold higher growth rate for the QAC-P2/QAC-P3 for 1/8 strains of *L. monocytogenes* (EGD) in ciprofloxacin (Table 2).

There were some significant strain × subpopulation interactions for an increase in *L. monocytogenes* growth rate in ciprofloxacin. For example, QAC-P1 of N1-227 had 4.0-fold increase in growth rate in ciprofloxacin, as compared to a 1.2- to 1.3-fold increase for QAC-P2 or QAC-P3 (Table 2).

### 3.3. Changes in the Survival of QAC-Adapted Subpopulations of L. monocytogenes in Ciprofloxacin-Containing Agar

Figure 4 shows the changes in survival (log CFU/mL) in the ciprofloxacin-containing agar model (2 µg/mL ciprofloxacin in TSAYE) for eight *L. monocytogenes* strains, before and after sublethal exposure to a fixed or gradually increasing concentration of QAC. Table 2 shows the fold changes in survival of three QAC-adapted subpopulations versus the non-adapted control for eight *L. monocytogenes* strains.

Two patterns were observed for changes in survival in ciprofloxacin-containing agar. (1) QAC-adapted subpopulations of the 4/8 *L. monocytogenes* strains (EGD, NRRL B 33155, NRRL B 33157, ATCC 43527) exhibited the highest increase in survival in ciprofloxacin, which was by 2.5 to 4.0 log CFU/mL for QAC-P1, by 3.2 to 4.1 log CFU/mL for QAC-P2, or by 2.2 to 4.0 log CFU/mL for QAC-P3 (*p* < 0.05), as compared to the non-adapted control (Figure 4A,E–G). This was equivalent to a percent increase in survival by 36.7 to 58.3S% for QAC-P1, or by 46.8 to 60.3% QAC-P2, or by 33.1 to 57.3% QAC-P3 in ciprofloxacin, as compared to the non-adapted control for these 4/8 strains (Table 2). In this group, the QAC-adapted subpopulations of NRRL B 33157 (serotype 4b) followed by EGD (serotype 1/2a) exhibited the highest survival increase in ciprofloxacin with some exceptions. (2) In contrast, QAC-adapted subpopulations of the 3/8 *L. monocytogenes* strains (N1-227, NRRL 33109 and ATCC 19116) showed no significant change in survival in ciprofloxacin, as compared to the non-adapted control.

With few exceptions within the six strains belonging to serotype 4b, QAC-adapted subpopulation NRRL B 33157 exhibited the highest percent increase in survival in ciprofloxacin, followed by serotype 1/2a (EGD), while that of serotype 4c (ATCC 19116) did not differ from the non-adapted control (Figure 4).

There were some significant strain × subpopulation interactions for an increase in survival in ciprofloxacin for 1/8 *L. monocytogenes* strains. For example, QAC-P2 of ScottA exhibited the highest percent survival increase in ciprofloxacin, as compared to QAC-P1 or QAC-P3.

When the ciprofloxacin level increased from 2 µg/mL to 4 µg/mL in agar, all QAC-adapted subpopulations were non-detectable for all eight strains of *L. monocytogenes*, similar to that of the non-adapted control (data not shown).

## 4. Discussion

The objective of this study was to evaluate the influence of sublethal concentrations of biocide QAC in inducing heterologous stress-response to fluoroquinolone antibiotic ciprofloxacin in *L. monocytogenes*, under controlled laboratory conditions. In the food processing environments, even though sanitizers and disinfectants were routinely used at 50–100 times greater than that of their minimum bactericidal concentration (MBC) to kill foodborne bacterial pathogens, planktonic cells or biofilms that might be present in the crevices might be frequently exposed to lower or sublethal concentrations of biocides because of dilution caused by the wetness on those surfaces or by interactions with food residues. Cleaning and disinfection steps might also leave residues of sanitizers or disinfectants, which might expose bacterial cells to a gradient of concentrations in the processing environments. Recent findings showed that such gradual exposure to sublethal concentrations of biocides could co-select for bacterial cells that were tolerant to lethal concentrations of biocides [10] or might cross-protect against antibiotics [24]. Therefore, it is important to understand the role of sublethal concentrations of biocides in the emergence of heterologous stress-response in *L. monocytogenes*, which might lead to antibiotic tolerance/resistance development.

We tested three approaches for creating the continuous exposure to sublethal concentrations of QAC against actively growing planktonic cells of *L. monocytogenes* and evaluated the subsequent changes in antibiotic susceptibility against ciprofloxacin by three different methods—(1) detecting the changes in short-range MIC of QAC-adapted subpopulations; (2) detecting the changes in the growth rate of QAC-adapted subpopulations in the ciprofloxacin-containing broth model; and (3) detecting the changes in survival of the QAC-adapted subpopulations in the ciprofloxacin-containing agar model. Our results showed that there was a potential for the development of low-level tolerance to ciprofloxacin in the *L. monocytogenes* strains after continuous sublethal exposure to QAC. A significant increase in the short-range MIC of the QAC-adapted subpopulations was observed to be about 1.5- to 2.9-fold for 4 out of 8 *L. monocytogenes* strains used in this study. A similar low-level 2–4 fold increase in MIC of ciprofloxacin was observed in *L. monocytogenes* after step-wise exposure to an increasing concentration of QAC, where 11 ciprofloxacin-sensitive strains with an initial MIC of 1 µg/mL, exhibited a 2-fold increase in MIC to 2 µg/mL of ciprofloxacin, after QAC adaptation [22]. While five other ciprofloxacin-sensitive *L. monocytogenes* strains with an MIC of 1 µg/mL exhibited a 4-fold increase in ciprofloxacin MIC to 4 µg/mL after QAC adaptation. In these and our study, the highest MIC increase of ciprofloxacin against QAC-adapted subpopulations of *L. monocytogenes* strains never exceeded beyond 8 µg/mL of ciprofloxacin. In other studies, a 4-fold increase in MIC of ciprofloxacin to 32 µg/mL against some *L. monocytogenes* strains was observed for the cells adapted to QAC [21]. A recent publication by Guerin et al., 2021 reported similar findings in which, repeated exposure to sublethal BC and DDAC resulted in a lower susceptibility to ciprofloxacin in 14 and 21 *L. monocytogenes* strains, respectively. BC exposed *L. monocytogenes* strains exhibited a 2.6-fold increase in MIC of ciprofloxacin from 2.1 µg/mL for control to 5.48 µg/mL in the adapted strains. DDAC-exposed cells exhibited a 4.5-fold increase in MIC from 2.1 µg/mL for the control to 9.48 µg/mL in the adapted strains. Additionally, these strains continued to exhibit reduced susceptibility to ciprofloxacin after the de-adaptation step [25]. Other pathogens such as *Pseudomonas aeruginosa*, which was enriched for 33 generations in QAC, demonstrated as high as 256-fold increase in MIC of ciprofloxacin [26].

While the increase in the short-range MIC of the QAC-adapted subpopulations of *L. monocytogenes* was statistically significant, these changes in the QAC-adapted subpopulations were below the standard breakpoints for ciprofloxacin. However, such increases in the short-range MIC of ciprofloxacin against QAC-adapted subpopulations of *L. monocytogenes* that were non-detectable by the standard MIC method should not be ignored, since the subpopulations of adapted cells were likely to be present to allow for faster growth rate under prolonged selective pressure. Eventually, such low-level MIC changes might potentially lead to a broader or high-level antibiotic resistance against ciprofloxacin [27]. MIC is an offline measurement tool that monitors the end-point results but is not intended for determining the microbial response continuously. Since a small increase in MIC remains usually undetected by the standard MIC [28], we employed growth kinetics to determine the dynamic effect of ciprofloxacin on the adapted cells of *L. monocytogenes*. QAC-adapted subpopulations of *L. monocytogenes* exhibited a significantly shorter lag phase and a faster growth rate in the presence of ciprofloxacin-containing broth. Similar findings with other bacteria led to understanding the role of the lag phase in the development of antibiotic tolerance. In general, the lag phase protects the bacteria from antibiotics, causing delayed or extended growth [29]. Such kinetic growth assays monitoring the duration of lag phases is an indicator of dose-dependent antibiotic inhibition [23]. The decrease in lag phase shown is an early sign of the competitive advantage of the adapted subpopulation over the non-tolerant subpopulation. By employing 1/2 MIC (2 µg/mL) of ciprofloxacin in broth, we observed the growth dynamics in the logarithmic phase of the adapted subpopulation over the non-adapted control. When ciprofloxacin concentration in broth increases or is equal to that of the MIC concentration, the growth dynamics of the adapted and non-adapted extend and takes longer. The two non-adapted strains (ATCC 19116 and NRRL-B-33109) that exhibited higher than 5 µg/mL of ciprofloxacin MIC were unaffected by this comparison and might require a much higher threshold concentration. Numerous factors affect the tolerance in bacteria, however, changes in the lag time were considered to be the first step towards the development of antibiotic tolerance. While we did not study the effect of higher concentration of ciprofloxacin in broth in decreasing the lag phase of the QAC-adapted subpopulations, the extension of any lag phase in the presence of lethal concentrations of antibiotics might actually prevent the killing of bacteria [30]. It appears that the kinetic growth curves of the QAC-adapted subpopulations of *L. monocytogenes* observed in our study indicate the possible presence of the subpopulations of actively growing adapted cells, in the presence of ciprofloxacin-containing broth.

Therefore, to separate the subpopulations of adapted cells versus non-adapted cells, the actively growing planktonic cells of the QAC-adapted subpopulations of *L. monocytogenes* were plated on the ciprofloxacin-containing agar model. A significant increase in survival of the QAC-adapted subpopulations of *L. monocytogenes* was observed in the presence of ciprofloxacin-containing agar. In general, the QAC-adapted subpopulations yielded about 2 to 4 log CFU/mL higher loads on ciprofloxacin-containing agar, as compared to the non-adapted cells for 4 out of the 8 strains tested. The representative individual colonies isolated on the ciprofloxacin-containing agar for the QAC-adapted subpopulations of *L. monocytogenes* need further study through whole-genome analysis.

Our results showed that, conversely to the other *L. monocytogenes* strains, the ciprofloxacin susceptibility of two strains ATCC 19116 and NRRL B-33109 was never altered by the different sublethal exposures to QAC in any subpopulation (QAC-P1, QAC-P2, or QAC-P3). We did not observe any increase in the short-range MIC, any increase in growth rate, and any increase in survival of the QAC-adapted subpopulations of these two strains, as compared to their non-adapted controls. This indicates that there might be two possibilities at work—(1) there might be a strain-specific response in QAC-adapted subpopulations of *L. monocytogenes* in the development of low-level tolerance against ciprofloxacin; and (2) there is an inability of some strains to adapt under certain conditions, perhaps requiring a longer exposure and higher sublethal shocks. By comparing their genomes with those of other strains that are able to adapt, it would be interesting to understand why both these strains did not adapt to QAC under some conditions. This would inform the physiological and genetic pathways that are potentially involved in adaptation to QAC and fluoroquinolone cross-resistance development in different *L. monocytogenes* strains.

Generally, in *L. monocytogenes*, the cross-resistance between antibiotics and QAC is linked to active efflux or due to acquisition of transferable genetic materials [17]. Resistance to fluoroquinolone is multifactorial and can be via one or a combination of target-site gene mutations, increased production of multidrug-resistance (MDR) efflux pumps, modifying enzymes, or target-protection proteins. The overexpression of regulatory genes that control the expression of native efflux pumps localized in the bacterial membrane such as *mdrL* and *lde* genes occurred only in the BAC-resistant strains [31]. *L. monocytogenes* strains harboring these genes or strains that are intrinsically or adaptively tolerant to biocides have a higher growth advantage over the ones that lack these parameters [32]. The efflux pumps *lde* and *mdrL* are responsible for demonstrating an increased tolerance to QAC and ciprofloxacin. The *lde* gene encodes for a multi-drug efflux pump, which was associated with fluoroquinolone resistance and was previously described in some strains of *L. monocytogenes* [33]. In a future study, we will explore if the expression of these efflux pump regulatory gene expression influence the low-level tolerance to ciprofloxacin in the QAC-adapted subpopulations of *L. monocytogenes* observed in this study.

Our continuing work will explore the following four aspects. (1) Using a pool of naturally occurring strains, we will compare the resistance level to ciprofloxacin *L. monocytogenes* and their corresponding level of resistance to QAC. (2) Using at least one strain of *L. monocytogenes*, we will perform WGS sequencing before and after adaptation to QAC and the selected gene expression associated with cross-tolerance to QAC and ciprofloxacin. (3) The stability of the acquired QAC-induced cross-tolerance to ciprofloxacin in the subpopulations of *L. monocytogenes*. (4) It would also be interesting to understand the correlation between both homologous and heterologous tolerance response in *L. monocytogenes* after QAC adaptation, with respect to ciprofloxacin and other selected antibiotics.

In conclusion, our findings show the potential for the development of low-level tolerance to fluoroquinolone antibiotic ciprofloxacin in QAC-adapted subpopulations of some *L. monocytogenes* strains that resulted in a significant increase in short-range MIC, growth rate, and survival in broth/agar models. Although we did not observe the development of higher level tolerance/resistance to ciprofloxacin in QAC-adapted subpopulations of *L. monocytogenes* in these eight strains tested, such low-level responses might have survival advantage in the gradual emergence of antibiotic-resistant subpopulations. Therefore, the early detection of low-level tolerance to antibiotics is vital, because such phenotypic variants might serve as a stepping-stone towards clinical antibiotic resistance development in the environmentally persisting stress-adapted *L. monocytogenes* strains, if subjected to cycles of sublethal exposure to QAC or other biocides in various environments.

## Figures and Tables

**Figure 1 microorganisms-09-01052-f001:**
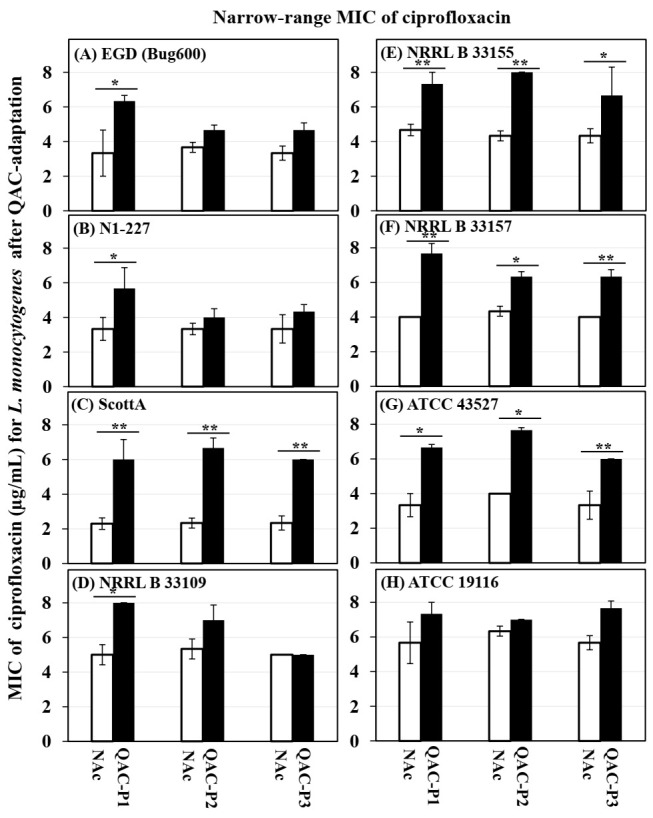
Changes in narrow-range MIC of ciprofloxacin (µg/mL) for three QAC-adapted subpopulations (closed bars) compared to non-adapted controls (NAc, open bars) of eight *L. monocytogenes* strains (**A**–**H**). Error bars indicate standard errors of means. Statistically significant *p* values are indicated by asterisks (* *p* < 0.05; ** *p* < 0.01;) that were obtained using the unpaired two-tailed *t*-test.

**Figure 2 microorganisms-09-01052-f002:**
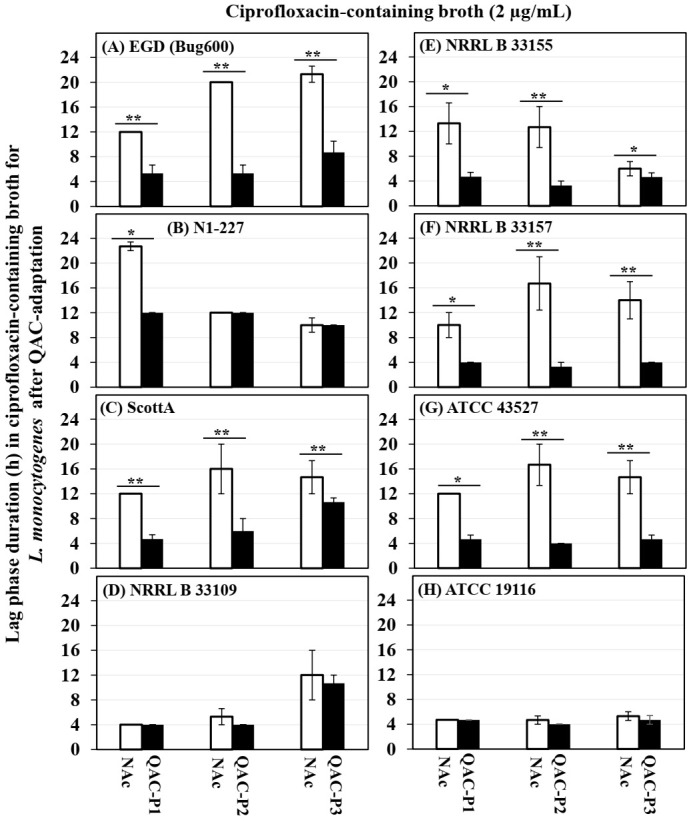
Changes in the lag phase duration (h) for the three QAC-adapted subpopulations (closed bars) in ciprofloxacin-containing broth (2 µg/mL) as compared to the non-adapted controls (NAc, open bars) of eight *L. monocytogenes* strains (**A**–**H**) at 37 °C. Error bars indicate the standard errors of means. Statistically significant *p*-values are indicated by asterisks (* *p* < 0.05; ** *p* < 0.01;) that were obtained using the unpaired two-tailed *t*-test.

**Figure 3 microorganisms-09-01052-f003:**
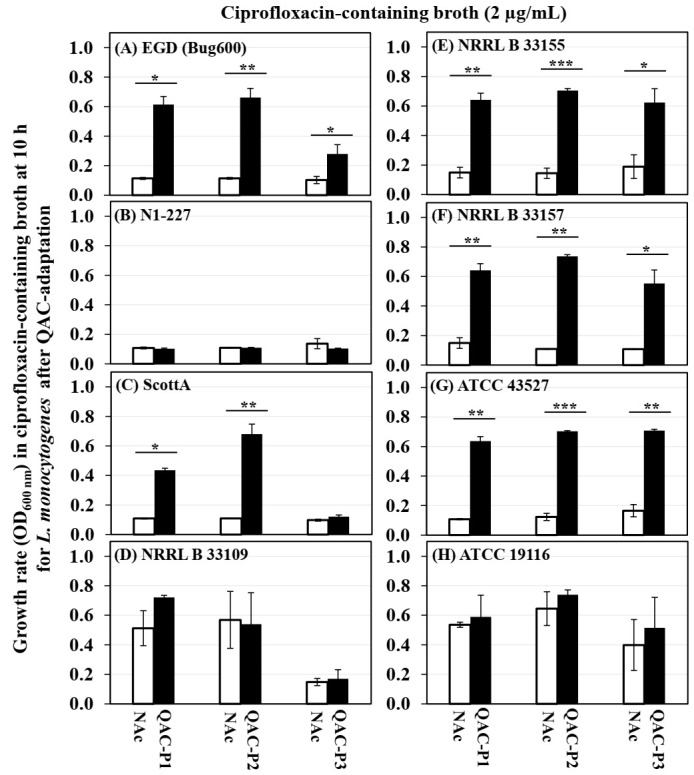
Changes in growth rate (OD_600_) for three QAC-adapted subpopulations (closed bars) in the ciprofloxacin-containing broth (2 µg/mL) at 10 h, as compared to the non-adapted control (NAc, open bars) of eight *L. monocytogenes* strains (**A**–**H**) at 37 °C. Error bars indicate the standard errors of means. Statistically significant *p*-values are indicated by the asterisks (* *p* < 0.05; ** *p* < 0.01; *** *p* < 0.001) that were obtained using the unpaired two-tailed *t*-test.

**Figure 4 microorganisms-09-01052-f004:**
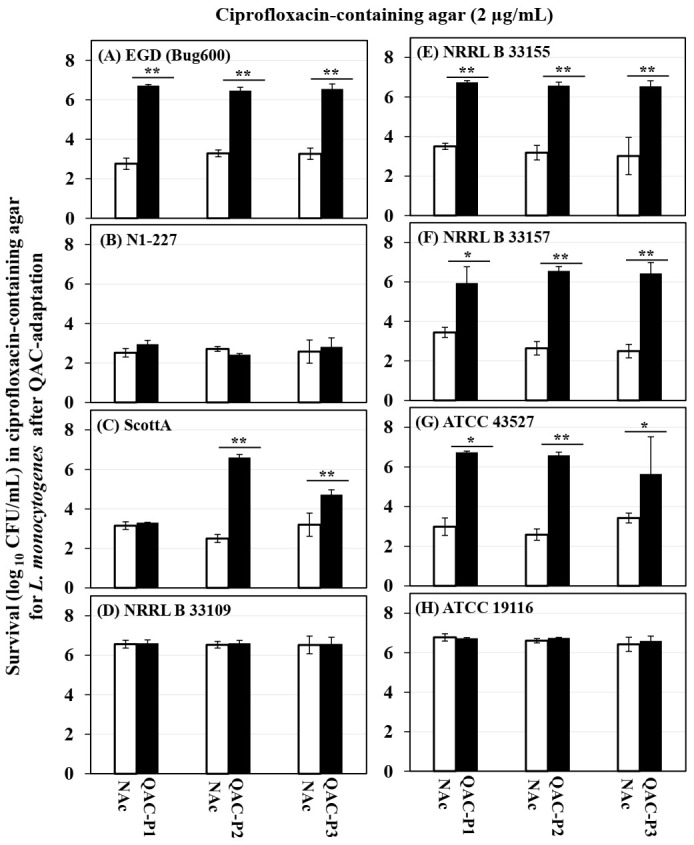
Changes in survival (log_10_ CFU/mL) for three QAC-adapted subpopulations (closed bars) in ciprofloxacin-containing agar (2 µg/mL) as compared to the non-adapted control (NAc, open bars) of eight *L. monocytogenes* strains (**A**–**H**). Error bars indicate the standard errors of means. Statistically significant *p*-values are indicated by asterisks (* *p* < 0.05; ** *p* < 0.01;) that were obtained using the unpaired two-tailed *t*-test.

**Table 1 microorganisms-09-01052-t001:** *L. monocytogenes* strains used in this study.

Species	Designation	Lineage	Serotype	Source	Isolation Source
*L. monocytogenes*	N1-227	I	4b	CDC, Atlanta	Food epidemic
*L. monocytogenes*	ATCC 19116	III	4c	University of Wisconsin	Poultry
*L. monocytogenes*	ScottA	I	4b	FDA	Human epidemic
*L. monocytogenes*	EGD (Bug600)	II	1/2a	Institute Pasteur	Guinea pigs
*L. monocytogenes*	NRRL B-33109	I	4b	USDA-ARS, NADC	Cooler condenser
*L. monocytogenes*	NRRL B-33155	I	4b	USDA-ARS, NADC	Sodium caesinate epidemic strain, CA, 1985 outbreak
*L. monocytogenes*	NRRL B-33157	I	4b	USDA-ARS, NADC	Insect debris found in cheese plant
*L. monocytogenes*	ATCC 43257	I	4b	CDC, Atlanta	Mexican Style cheese, CA

**Table 2 microorganisms-09-01052-t002:** Fold increase in MIC of ciprofloxacin, fold increase in growth rate (OD_600_ in ciprofloxacin-containing broth), and percent increase in survival (in ciprofloxacin-containing agar) of three QAC-adapted subpopulations (QAC-P1, QAC-P2, and QAC-P3), compared to non-adapted control of eight *L. monocytogenes* strains.

*Listeria monocytogenes* Strains	Fold Increase ^1^ in MIC of Ciprofloxacin for QAC-P1 ± SE	Fold Increase ^2^ in Growth (OD_600_) of QAC-P1 Compared to Non-Adapted Control in Ciprofloxacin-Containing Broth at Different Time Points				Percentage Increase ^3^ in Survivals in Ciprofloxacin-Agar for QAC-P1 ± SE
		10 h	12 h	20 h	24 h	
EGD (Bug600)	1.9 ± 0.2 ^a^	4.5 ± 0.8 ^a^	5.3 ± 1.0 ^a^	2.4 ± 0.4 ^a^	1.9 ± 0.4 ^b^	58.3 ± 5.4 ^a^
N1 227	1.7 ± 0.3 ^b^	1.0 ± 0.0 ^c^	0.9± 0.0 ^b^	2.4 ± 0.5 ^a^	4.0 ± 0.2 ^a^	5.77 ± 0.5 ^c^
Scott A	2.6 ± 0.3 ^a^	4.0 ± 0.2 ^a^	4.8 ± 0.2 ^a^	3.1 ± 0.9 ^a^	2.2 ± 1.0 ^a^	1.75 ± 2.7 ^c^
NRRL B 33109	1.6 ± 0.2 ^b^	1.4 ± 0.4 ^b^	1.4 ± 0.5 ^b^	1.2 ± 0.3 ^b^	1.1 ± 0.1 ^d^	1.20 ± 0.3 ^c^
NRRL B 33155	1.6 ± 0.2 ^b^	3.9 ± 0.7 ^a^	3.4 ± 1.3 ^a^	2.3 ± 1.1 ^a^	1.5 ± 0.2 ^c^	46.4 ± 1.7 ^a^
NRRL B 33157	1.9 ± 0.1 ^a^	3.6 ± 0.4 ^a^	4.3 ± 0.8 ^a^	1.0 ± 0.1 ^b^	1.0 ± 0.1 ^d^	36.7 ± 13.6 ^b^
ATCC 43257	2.0 ± 0.4 ^a^	5.0 ± 1.1 ^a^	5.9 ± 0.2 ^a^	2.5 ± 0.4 ^a^	2.2 ± 0.4 ^b^	55.6 ± 10.6 ^a^
ATCC 19116	1.3 ± 0.1 ^b^	1.1 ± 0.3 ^c^	1.1 ± 0.2 ^b^	1.2 ± 0.0 ^b^	1.1 ± 0.1 ^d^	0.80 ± 0.4 ^c^
***Listeria monocytogenes* Strains**	**Fold Increase ^1^ in MIC of Ciprofloxacin for QAC-P2 ± SE**	**Fold Increase ^2^ in Growth (OD_600_) of QAC-P2 Compared to Non-Adapted Control in Ciprofloxacin-Containing Broth at Different Time Points**				**Percentage Increase ^3^ in Survivals in Ciprofloxacin-Agar for QAC-P2 ± SE**
		**10 h**	**12 h**	**20 h**	**24 h**	
EGD (Bug600)	1.3 ± 0.1 ^c^	4.9 ± 0.9 ^a^	5.6 ± 1.0 ^a^	5.1 ± 0.4 ^a^	2.1 ± 0.4 ^a^	46.8 ± 4.1 ^b^
N1 227	1.2 ± 0.3 ^c^	1.0 ± 0.0 ^b^	1.0 ± 0.0 ^c^	1.0 ± 0.5 ^b^	1.3 ± 0.2 ^a^	−4.6 ± 2.3 ^d^
Scott A	2.9 ± 0.1 ^a^	5.1 ± 0.2 ^a^	5.6 ± 0.2 ^a^	3.7 ± 0.9 ^a^	1.6 ± 1.0 ^a^	60.3 ± 2.6 ^a^
NRRL B 33109	1.3 ± 0.0 ^c^	0.9 ± 0.4 ^b^	0.8 ± 0.5 ^c^	0.9 ± 0.3 ^b^	1.0 ± 0.1 ^b^	0.90 ± 0.8 ^c^
NRRL B 33155	1.8 ± 0.1 ^b^	3.3 ± 0.7 ^a^	2.6 ± 1.3 ^b^	2.0 ± 1.1 ^a^	1.2 ± 0.2 ^a^	49.8 ± 7.1 ^b^
NRRL B 33157	1.5 ± 0.0 ^b^	6.8 ± 0.4 ^a^	5.7 ± 0.8 ^a^	2.2 ± 0.1 ^a^	1.2 ± 0.1 ^a^	58.9 ± 3.9 ^a^
ATCC 43257	1.9 ± 0.1 ^b^	6.7 ± 1.1 ^a^	5.1 ± 0.2 ^a^	3.1 ± 0.4 ^a^	2.1 ± 0.4 ^a^	59.7 ± 4.2 ^a^
ATCC 19116	1.2 ± 0.1 ^c^	1.1 ± 0.3 ^b^	1.0 ± 0.2 ^c^	1.0 ± 0.0 ^b^	1.0 ± 0.1 ^b^	2.85 ± 1.7 ^c^
***Listeria monocytogenes* Strains**	**Fold Increase ^1^ in MIC of Ciprofloxacin for QAC-P3 ± SE**	**Fold Increase ^2^ in Growth (OD_600_) of QAC-P3 Compared to Non-Adapted Control in Ciprofloxacin-Containing Broth at Different Time Points**				**Percentage Increase ^3^ in Survivals in Ciprofloxacin-Agar for QAC-P3 ± SE**
		**10 h**	**12 h**	**20 h**	**24 h**	
EGD (Bug600)	1.4 ± 0.1 ^b^	2.1 ± 0.6 ^b^	3.1 ± 1.1 ^b^	3.4 ± 0.7 ^a^	2.3 ± 1.0 ^a^	48.1 ± 1.9 ^a^
N1 227	1.3 ± 0.3 ^b^	0.8 ± 0.2 ^c^	1.2 ± 0.4 ^c^	1.3 ± 0.4 ^b^	1.2 ± 0.1 ^b^	3.3 ± 5.4 ^d^
Scott A	2.6 ± 0.3 ^a^	1.2 ± 0.2 ^c^	1.8 ± 0.6 ^c^	2.0 ± 0.3 ^a^	1.2 ± 0.3 ^b^	22.6 ± 6.8 ^c^
NRRL B 33109	1.0 ± 0.0 ^c^	1.1 ± 0.8 ^c^	0.8 ± 1.7 ^c^	1.2 ± 1.5 ^b^	1.1 ± 0.4 ^b^	0.41 ± 1.3 ^d^
NRRL B 33155	1.5 ± 0.3 ^a^	1.7 ± 1.0 ^c^	2.2 ± 1.5 ^b^	1.6 ± 1.4 ^a^	1.2 ± 0.1 ^b^	52.1 ± 5.0 ^a^
NRRL B 33157	1.6 ± 0.1 ^b^	5.1 ± 0.7 ^a^	5.2 ± 0.7 ^a^	1.3 ± 1.4 ^b^	1.0 ± 0.0 ^b^	57.3 ± 7.5 ^a^
ATCC 43257	1.8 ± 0.5 ^a^	6.4 ± 0.7 ^a^	6.6 ± 0.6 ^a^	3.5 ± 1.0 ^a^	1.9 ± 0.2 ^a^	33.1 ± 17.6 ^b^
ATCC 19116	1.4 ± 0.1 ^b^	1.3 ± 2.5 ^c^	1.1 ± 2.5 ^c^	0.9 ± 0.2 ^b^	0.9 ± 0.1 ^b^	3.24 ± 1.1 ^d^

Different letter indicate significant differences (*p* < 0.05) by Duncan’s multiple range test. for **^1^** Fold Increase in MIC of Ciprofloxacin for QAC-Adapted Subpopulations, or **^2^** Fold Increase in Growth (OD600) of QAC-Adapted Subpopulations Compared to Non-Adapted Control in Ciprofloxacin-Containing Broth at Different Time Points, or **^3^** Percentage Increase in Survivals in Ciprofloxacin-Agar for QAC-Adapted Subpopulations.

## Data Availability

All relevant data are within the paper.

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
