# Peer review of "Low-Level Tolerance to Fluoroquinolone Antibiotic Ciprofloxacin in QAC-Adapted Subpopulations of Listeria monocytogenes"

_microorganisms, 2021, doi:10.3390/microorganisms9051052_

Round 1

Reviewer 1 Report

In the submitted manuscript, Kode et. al. investigated the impact of Benzalkonium chloride exposure on the ciprofloxacin susceptibility of 8 Listeria monocytogenes strains. The authors reported the development of a low-level tolerance to ciprofloxacin after sublethal adpatation to BC in 4-5 strains depending on method of exposition . Although the study still descriptive and not adressed the mechanisms underlying adaptation to QAC and ciprofloxacin resistance development, the work is well conducted and clearly written. It should provide valuable information to readers in the field who are interested in this topic. Some suggestions for the authors to consider:

-Results showed that, conversely to the other Listeria strains,  the ciprofloxacin susceptibility of strains ATCC 19116 and B-33109 was never altered by the different exposition to QAC (QAC-P1, P2 or P3). This point is interesting and not sufficiently discussed from my point of view. It will be interesting to understand why both these strains did not adapt to QAC by comparing their genomes with those of strains able to adapt. That would inform on the genetic pathways potentially involved in adaptation to QAC and fluoroquinolone cross-resistance development.

-When looking Supplementary data,  QAC-P3 adapted growth curves of NRRL B-33109 seemed significantly different from unadapted strain although that not appear on figure 3.

-It could be interesting to know the MIC of BC in adapted strains to appreciate the range of adaptation to the biocide. Is the ciprofloxacin MIC increase is systematically linked to a decrease of susceptibility to QAC?

-A very recent paper showed similar findings and should be added to the introduction or discussion section: PMID: 33670643

Reviewer 1 Report

Authors Responses

In the submitted manuscript, Kode et. al. investigated the impact of Benzalkonium chloride exposure on the ciprofloxacin susceptibility of 8 Listeria monocytogenes strains. The authors reported the development of a low-level tolerance to ciprofloxacin after sublethal adaptation to BC in 4-5 strains depending on method of exposition. Although the study still descriptive and not addressed the mechanisms underlying adaptation to QAC and ciprofloxacin resistance development, the work is well conducted and clearly written. It should provide valuable information to readers in the field who are interested in this topic. Some suggestions for the authors to consider:

-Results showed that, conversely to the other Listeria strains, the ciprofloxacin susceptibility of strains ATCC 19116 and B-33109 was never altered by the different exposition to QAC (QAC-P1, P2 or P3). This point is interesting and not sufficiently discussed from my point of view. It will be interesting to understand why both these strains did not adapt to QAC by comparing their genomes with those of strains able to adapt. That would inform on the genetic pathways potentially involved in adaptation to QAC and fluoroquinolone cross-resistance development.

Author’s response: As recommended by the reviewer, new discussion is added on this point in the revised manuscript using comments from the reviewer.

Our results showed that, conversely to the other L. monocytogenes strains, the ciprofloxacin susceptibility of two strains ATCC 19116 and NRRL B-33109 was never altered by the different sublethal exposures to QAC in any subpopulation (QAC-P1, QAC-P2 or QAC-P3). We did not observe any increase in short-range MIC, any increase in growth rate and any increase in survival of QAC-adapted subpopulations of these two strains compared to their non-adapted controls. This indicates there may be two possibilities at work: (1) there may be a strain-specific response in QAC-adapted subpopulations of L. monocytogenes in the development of low-level tolerance against ciprofloxacin; and (2) there is the inability of some strains to adapt under certain conditions perhaps requiring a longer exposure and higher sublethal shocks. By comparing their genomes with those of other strains that are able to adapt, it will be interesting to understand why both these strains did not adapt to QAC under some conditions. That would inform the physiological and genetic pathways potentially involved in adaptation to QAC and fluoroquinolone cross-resistance development in different L. monocytogens strains.

-When looking Supplementary data,  QAC-P3 adapted growth curves of NRRL B-33109 seemed significantly different from unadapted strain although that not appear on figure 3.

Author’s response:  There are no changes in Figure 3. We found one error and needed to update Supplement Figure S3. When checked that supplement figure, it was found that subfigure S3-D and S3-E are the same. We have submitted the updated Supplement Figure S3 with correct subfigure S3-D. This dataset matches with the original results for Figure 3 and the Supplement Figure S3.

-It could be interesting to know the MIC of BC in adapted strains to appreciate the range of adaptation to the biocide. Is the ciprofloxacin MIC increase is systematically linked to a decrease of susceptibility to QAC?

Author’s response: As part of our future work, we will explore the correlation between both homologous and heterologous tolerance in L. monocytogenes after QAC adaptation with respect to selected antibiotics. We will determine this relationship between QAC tolerance versus ciprofloxacin tolerance based on the changes in MIC of BAC in the adapted strains. This point is included in the revised manuscript.

-A very recent paper showed similar findings and should be added to the introduction or discussion section: PMID: 33670643

Author’s response: Guerin et al., 2021 reference is now included and discussed in the revised manuscript.

A recent publication by Guerin et al., 2021 reported similar findings in which by repeated exposure to sublethal BC and DDAC resulted in a lower susceptibility to ciprofloxacin in 14 and 21 L. monocytogenes strains, respectively. BC exposed L. monocytogenes strains exhibited a 2.6 fold increase in MIC of ciprofloxacin from 2.1 µg/ml for control to 5.48 µg/ml in the adapted strains.  DDAC exposed cells exhibited a 4.5 fold increase in MIC from 2.1 µg/ml for the control to 9.48 µg/ml in the adapted strains. Also, these strains continued to exhibit reduced susceptibility to ciprofloxacin after the de-adaptation step (Guérin et al., 2021).

We are grateful for the critical review of our manuscript. We are thankful to the reviewer for the excellent comments and suggestions on this manuscript.

 Ramakrishna Nannapaneni

Submission Date 7 May 2021

Reviewer 2 Report

The manuscript raises a very important problem related to the cross-acquisition of resistance to disinfectants and antibiotics by microorganisms. It is a relatively frequent and inevitable phenomenon, and at the same time researches in this aspect are still needed. The manuscript is carefully written. Most of the methods are correctly selected and described, and the results are clearly presented. Nevertheless, there are a few drawbacks and doubts that should be removed or cleared before publication of the manuscript:

  1. Why, when subculturing strains from freezing, the authors decided to use a selective medium instead of a multiplication medium, such as TSB, BHI, etc.?
  2. In Europe, there is a standard EN1276: 2010, with subsequent revisions, which specifies that hard water composed of solution A (19.84 g MgCl2 and 46.24 g CaCl2 was dissolved in 1000 mL H2O), B (35.02 g NaHCO3 dissolved in 1000 mL H2O) and sterile tap water in appropriate proportions is used to evaluate the effectiveness of disinfectants. Are there no such recommendations in the US and distilled water allowed?
  3. Why was TSBYE used instead of MHB, which is recommended in this type of research, when generating QAC resistance?
  4. I would not use the term "phenotype" in points 2.4-2.6. Phenotype means a set of body traits, including not only morphology, but also physiological properties, such as resistance to antibiotics. As a result of the experiment, all strains acquired resistance to QAC, i.e. they belonged to the same phenotype, although it was achieved by different methods. Therefore, I would use the term „group” or „subpopulation”.
  5. I recommend that the Authors check the stability of acquired QAC resistance, g. by carrying out a few passages on liquid or solid media without the addition of QAC and then reassessing QAC resistance in the passaged strains.
  6. It is worth checking the resistance to ciprofloxacin among several strains, which already, at the time of isolation from the environment, were characterized by a some level of resistance to QAC.
  7. I recommend performing WGS sequencing for at least one strain in "version" before and after generating QAC resistance, or at least selecting and evaluating gene expression that may be associated with cross resistance to QAC and ciprofloxacin.

Reviewer 2 Report

Authors Responses

The manuscript raises a very important problem related to the cross-acquisition of resistance to disinfectants and antibiotics by microorganisms. It is a relatively frequent and inevitable phenomenon, and at the same time researches in this aspect are still needed. The manuscript is carefully written. Most of the methods are correctly selected and described, and the results are clearly presented. Nevertheless, there are a few drawbacks and doubts that should be removed or cleared before publication of the manuscript:

Why, when subculturing strains from freezing, the authors decided to use a selective medium instead of a multiplication medium, such as TSB, BHI, etc.?

Author’s response: Although the frozen stocks are pure, our standard lab protocol is to maintain the refrigerated working stocks of all L. monocytogenes strains on a selective PALCAM agar for daily use in preparing the overnight cultures in TSBYE for growth and multiplication needs. This practice double-checked the purity of the frozen culture during the revival process and eliminated the cross contamination by other foodborne bacterial pathogens that we routinely use in the same lab.

In Europe, there is a standard EN1276: 2010, with subsequent revisions, which specifies that hard water composed of solution A (19.84 g MgCl2 and 46.24 g CaCl2 was dissolved in 1000 mL H2O), B (35.02 g NaHCO3 dissolved in 1000 mL H2O) and sterile tap water in appropriate proportions is used to evaluate the effectiveness of disinfectants. Are there no such recommendations in the US and distilled water allowed?

Author’s Response:  EPA SOP Number MB-30-02 describes procedures for the preparation of hard water. The EPA allows 500 ppm of hard water per gallon or standard tap water for dilution in preparing disinfectants. However, to eliminate any interferences and since all non-adapted controls are grown in TSBYE containing distilled water as a diluent for MIC and growth curves, we kept the same diluent in preparing QAC concentrations for comparing the adapted subpopulations for homologous and heterologous responses. We found that most researchers have diluted QAC in distilled water or deionized water or in broth like TSB/BHI/LB. For example, Guerin et al., 2021 used sterile water for MIC determination and did not state whether it was distilled/deionized /hard water. To et al. used BAC prepared in deionized water. Yu et al. used sterile distilled water for preparing QAC stock solution. We will use hard water in preparing the disinfectants in our further work.

Why was TSBYE used instead of MHB, which is recommended in this type of research, when generating QAC resistance?

Author’s response: TSB is a complex nutrient-rich general-purpose medium while MHB is commonly used for MIC testing of non-fastidious organisms. However, with our sequential disinfectant exposure, we obtained a subpopulation of L. monocytogenes which may need fastidious media for its growth. Such differences were noticeable since growth rates of both adapted and non-adapted subpopulations were slower in MHB versus TSB. Therefore, we used TSB supplemented with yeast extract for comparing the growth dynamics of QAC-adapted subpopulations with non-adapted control. Also, other researchers kept the same growth medium for the adaptation in disinfectants and for the subsequent cross-tolerance studies.

I would not use the term "phenotype" in points 2.4-2.6. Phenotype means a set of body traits, including not only morphology, but also physiological properties, such as resistance to antibiotics. As a result of the experiment, all strains acquired resistance to QAC, i.e. they belonged to the same phenotype, although it was achieved by different methods. Therefore, I would use the term „group” or „subpopulation”.

 Author’s response: As recommended by the reviewer, the term ‘phenotype(s)’ has been changed to ‘subpopulation(s)’ throughout the manuscript.

I recommend that the Authors check the stability of acquired QAC resistance, g. by carrying out a few passages on liquid or solid media without the addition of QAC and then reassessing QAC resistance in the passaged strains.

Author’s Response:  The stability of the acquired QAC resistance in the subpopulations of L. monocytogenes and how it differs with strains will be explored in our continuing work. This point is included in the discussion section of the revised manuscript.

It is worth checking the resistance to ciprofloxacin among several strains, which already, at the time of isolation from the environment, were characterized by a some level of resistance to QAC.

Author’s Response:  Using a pool of naturally occurring strains, we will compare the resistance level to ciprofloxacin L. monocytogenes and their corresponding level of resistance to QAC in our further work. This point is included in the discussion section of the revised manuscript.

I recommend performing WGS sequencing for at least one strain in "version" before and after generating QAC resistance, or at least selecting and evaluating gene expression that may be associated with cross resistance to QAC and ciprofloxacin.

Author’s Response:  Using at least one strain of L. monocytogenes, we will perform WGS sequencing before and after adaptation to QAC and will determine the selected gene expression associated with cross resistance to QAC and ciprofloxacin in our further work.  This point is included in the discussion section of the revised manuscript.

We are grateful for the critical review of our manuscript. We are thankful to the reviewer for the excellent comments and suggestions on this manuscript.

Ramakrishna Nannapaneni, Corresponding Author

Submission Date 7 May 2021

Reviewer 3 Report

The authors submitted a well-written manuscript addressing a critical antibiotic-resistant question where they found that 4/8 QAC adapted L. monocytogenes strains increased MIC for ciprofloxacin. The authors reported data on their three rationales to understand the short-range MIC, growth kinetics in broth, and solid agar media. The significance of the study is that they identified a key feature of QAC adaptation that might help early detection of high tolerant antibiotic-resistant pathogens. The study lacks mechanistic data with regards to the QAC adaptation. It would be interesting to understand the genetic regulation of different genes and pathways contributing to the enhanced tolerance.

Major comments:

1. Figure 1- At what time point was the MIC recorded? Based on table 2, there are three-time points. (10h, 12h, 20h, 24h). But the method mentioned as 48hr incubation. An interesting observation indicates that over time the growth decreases. What did the authors think about that?

In Table 2, what does the number 2 and 3 associated with fold increase and % increase meant? Sorry, I could not find the citation legend.

2. Figure 2- What was the rationale for checking the lag phase duration? In fig 1, most of the strains had MIC more than 5ug/ml, but in figure 2, the QAC adapted strains do not match the control even at 2ug/ml? Surprisingly, the authors found no growth at 4ug/ml.

3. Figure 3- At 10 hr, 5/8 had increased OD for QAC adapted to control. Did the authors attempt to check the morphology of the bacteria at 10hr? Why didn't the authors use other colorimetric assays to confirm the kinetics or MIC?

4. Figure 4- Interestingly, both NRRL B 33109 and ATCC 19116 (2/8) had a very different pattern for CFU/ml for the control. What could be the reason that the growth rate for both the strains is almost the same in control and QAC adapted?

*If 4/8 strains exhibit increased tolerance to antibiotics upon QAC treatment, what factors should be considered for the study's significance?

Reviewer 3 Report

Authors Responses

The authors submitted a well-written manuscript addressing a critical antibiotic-resistant question where they found that 4/8 QAC adapted L. monocytogenes strains increased MIC for ciprofloxacin. The authors reported data on their three rationales to understand the short-range MIC, growth kinetics in broth, and solid agar media. The significance of the study is that they identified a key feature of QAC adaptation that might help early detection of high tolerant antibiotic-resistant pathogens. The study lacks mechanistic data with regards to the QAC adaptation. It would be interesting to understand the genetic regulation of different genes and pathways contributing to the enhanced tolerance.

Major comments:

  1. Figure 1- At what time point was the MIC recorded? Based on table 2, there are three-time points. (10h, 12h, 20h, 24h). But the method mentioned as 48hr incubation. An interesting observation indicates that over time the growth decreases. What did the authors think about that?

Author’s response: While MIC of ciprofloxacin was recorded after 48 h, the earlier four time points given in Table 2 (10 h, 12 h, 20 h and 24 h) illustrated the dynamics of growth of QAC-adapted and non-adapted L. monocytogenes. These four early time points were chosen from the growth curves given in supplement. By observing the fold increase in the growth rate as summarized in Table 2, we were able to pin-point the survival advantage of QAC-adapted subpopulations in ciprofloxacin containing broth compared to the non-adapted control. The QAC-adapted subpopulations showed characteristic growth advantage in the logarithmic phase (10-12 h) prior to their growth cycle ends by nutrient limitation. When nutrients depleted at the end of the cycle, the growth rate (or OD) of both adapted and non-adapted cells decreased due to the onset of death phase.

In Table 2, what does the number 2 and 3 associated with fold increase and % increase meant? Sorry, I could not find the citation legend.

Author’s response: In Table 2, the superscripts numbers 1, 2 and 3 are now updated in the citation legend in the footnote below.

  1. Figure 2- What was the rationale for checking the lag phase duration? In fig 1, most of the strains had MIC more than 5ug/ml, but in figure 2, the QAC adapted strains do not match the control even at 2ug/ml? Surprisingly, the authors found no growth at 4ug/ml.

Author’s response: The lag phase can be an important factor in evaluating the tolerance response in bacteria towards antibiotics which allows the cells to endure antibiotic stress (Li et al., 2016). The decrease in lag phase shown in Figure 2 is an early sign of the competitive advantage of the adapted subpopulation over the non-tolerant subpopulation. By employing 1/2 MIC (2 µg/ml) of ciprofloxacin in broth, we observed the growth dynamics in the logarithmic phase of the adapted subpopulation over the non-adapted control. When ciprofloxacin concentration in broth increases or equal to that of the MIC concentration, the growth dynamics of adapted and non-adapted extend and takes longer. The two non-adapted strains (ATCC 19116 and NRRL-B-33109) that exhibited higher than 5 µg/ml of ciprofloxacin MIC are unaffected by this comparison and may require a higher threshold concentration.  These points are included in the discussion section of the revised manuscript.

  1. Figure 3- At 10 hr, 5/8 had increased OD for QAC adapted to control. Did the authors attempt to check the morphology of the bacteria at 10hr? Why didn't the authors use other colorimetric assays to confirm the kinetics or MIC?

Author’s response: While our primary focus was on establishing the growth dynamics and quantitative survival patterns of QAC-adapted subpopulations, we will explore the morphological changes in our further work. We have previously published morphological changes in L. monocytogenes in chlorine adapted cells and such experiments will be conducted with QAC and other disinfectants by SEM and TEM. Since the OD600nm based turbidity assay is the fastest and most widely used method for observing the differences in growth rate between stress adapted and non-adapted subpopulations and when supported by survival counts in ciprofloxacin containing agar in our experiments, we did not depend on other calorimetric assays.

  1. Figure 4- Interestingly, both NRRL B 33109 and ATCC 19116 (2/8) had a very different pattern for CFU/ml for the control. What could be the reason that the growth rate for both the strains is almost the same in control and QAC adapted?

Author’s response: The two non-adapted strains (ATCC 19116 and NRRL-B-33109) which are unaffected may require much higher threshold stress levels for changes in growth dynamics of adapted over non-adapted subpopulations to be tested in our further work. This point is included in the discussion section of the revised manuscript.

*If 4/8 strains exhibit increased tolerance to antibiotics upon QAC treatment, what factors should be considered for the study's significance?

Author’s response: We are exploring these factors: (1) there may be a strain-specific response in QAC-adapted subpopulations of L. monocytogenes in the development of low-level tolerance against ciprofloxacin; and (2) those strains with the inability to adapt under certain experimental conditions perhaps will require a longer exposure time and higher sublethal shocks. These points are included in the discussion section of the revised manuscript.

We are grateful for the critical review of our manuscript. We are thankful to the reviewer for the excellent comments and suggestions on this manuscript.

Ramakrishna Nannapaneni, Corresponding Author

Submission Date 7 May 2021